# The Role of mTOR in B Cell Lymphoid Malignancies: Biologic and Therapeutic Aspects

**DOI:** 10.3390/ijms241814110

**Published:** 2023-09-14

**Authors:** Eleni A. Karatrasoglou, Maria Dimou, Alexia Piperidou, Eleftheria Lakiotaki, Penelope Korkolopoulou, Theodoros P. Vassilakopoulos

**Affiliations:** 1First Department of Pathology, National and Kapodistrian University of Athens, Laikon General Hospital, 15773 Athens, Greece; ellakiotaki@gmail.com (E.L.); pkorkol@med.uoa.gr (P.K.); 2Department of Hematology and Bone Marrow Transplantation, National and Kapodistrian University of Athens, Laikon General Hospital, 15773 Athens, Greece; msdimou@gmail.com (M.D.); alexia_piper@hotmail.com (A.P.); theopvass@hotmail.com (T.P.V.)

**Keywords:** mTOR, DLBCL, MCL, mTOR inhibitors, temsirolimus, everolimus, therapy

## Abstract

Non-Hodgkin lymphoma’s (NHL) incidence is rising over time, and B cell lymphomas comprise the majority of lymphomas. The phosphoinositide 3-kinase (PI3K)/v-akt murine thymoma viral oncogene homologue 1 (Akt)/mammalian target of the rapamycin (mTOR) signaling pathway plays a critical role in a variety of cellular processes, such as cell proliferation and survival. Its role in lymphomagenesis is confirmed in many different types of B cell lymphomas. This review is mainly focused on the PI3K/v-akt/mTOR pathway-related oncogenic mechanisms in B cell NHLs with an emphasis on common B cell lymphoma types [diffuse large B cell lymphoma (DLBCL) and mantle cell lymphoma (MCL)]. Furthermore, it summarizes the literature regarding the clinical applications of the mTOR inhibitors temsirolimus and everolimus in B cell NHLs, which have been tested in a range of clinical trials enrolling patients with B cell malignancies, either as monotherapy or in combination with other agents or regimens.

## 1. Introduction

The incidence of NHLs has increased steadily over the past few decades, with an estimated 545,000 new cases and 260,000 deaths of NHL globally in 2020 [1] and B cell lymphoma accounting for 95% of the lymphomas [2]. The activation of the PI3-K/Akt/mTOR signaling pathway is well established in various types of B cell NHLs, such as DLBCL, MCL, chronic lymphocytic leukemia (CLL) and hairy cell leukemia (HCL) [3]. The need for more effective therapies with fewer side effects has rendered the use of mTOR inhibitors, temsirolimus and everolimus, an attractive therapeutic solution either as monotherapy or in combination with other drugs.

## 2. The PI3K/v-akt/mTOR Pathway

The mTOR signaling pathway plays a critical role in a variety of cellular processes, such as cell proliferation and survival.

### 2.1. PI3Ks

PI3Ks belong to a family of lipid kinases that can be classified into three classes (named I–III) according to their specificity and structure [4]. PI3K class I consists of two different subgroups, including classes IA and IB. Class IA PI3Ks are activated by the receptor tyrosine kinases (RTK) growth factor via interaction with the Gβγ and Ras proteins, encoding for the catalytic subunit of the p110α, p110β and p110δ isoforms. Class IA regulatory genes are *PIK3R1*, *PIK3R2* and *PIK3R3*, encoding for the catalytic isoforms p110 (α, β, δ), *PIK3CA*, *PIK3CB* and *PIK3CD*, respectively. Class IB PI3Ks are activated by RTKs and code for the catalytic subunit p110γ [4,5,6]. While catalytic isoforms p110α and p110β are found in all cell types, p110δ and p110γ are exclusively restricted to hematopoietic cells. The PI3Ks class II includes three isoforms, namely *PI3KC2α, PI3KC2β* and *PI3KC2γ*, encoded by the *PIK3C2A, PIK3C2B* and *PIK3C2G* genes. This specific class has a single catalytic unit interacting directly with phosphorylated adapter proteins and lacks regulatory subunits [4,5]. The PI3Ks class III is encoded by the *PIK3C3* gene and consists of a catalytic hVps34 and a regulatory p150 subunit [4]. The main function of PI3K is the phosphorylation of the D3 position of phosphoinositide, converting the class I phosphorylate phosphatidylinositol-(4,5)-bisphosphate (PIP2) into phosphatidylinositol-(3,4,5)-trisphosphate (PIP3) [7,8]. The tumor suppressor phosphatase and tensin homologue (PTEN) antagonizes the PI3K by dephosphorylating PIP3 to PIP2. Mutations or deletions resulting in PTEN loss are frequent events in human cancers.

Over the past two decades, various approaches have led to a comprehensive view of the PTEN’s roles in hematopoiesis and leukemia/lymphoma development. The PTEN negatively regulates neural stem cell self-renewal by modulating its G0–G1 cell cycle entry and growth factor dependency [9]. PTEN loss indeed drives long-term hematopoietic stem cells (LT-HSCs) to enter the cell cycle, leave bone marrow and accumulate in the spleen. PTEN-null HSCs lose their ability for long-term reconstitution after transplantation and are eventually depleted. Mechanistically, mTOR activation can induce high rates of protein synthesis, which contributes to impaired PTEN-null HSC self-renewal [10].

### 2.2. Akt

Akt, a serine-threonine kinase also known as the protein kinase B, belongs to a wide variety of proteins initiated by PIP3 and represents one of the major oncogenic effectors of the PI3K/Akt pathway [4,11]. Akt kinase is involved in the regulation of cell cycle progression, differentiation, transcription, translation and angiogenesis while it exerts an antiapoptotic effect by phosphorylating and inactivating proapoptotic factors, such as BAD, caspase 9, NF-κB and GSK-3 [12,13,14,15,16]. Akt also regulates cyclinD stability and inhibits, via phosphorylation, the function of a negative regulator of the cell cycle, p27 [17,18]. Furthermore, the Akt kinase enhances telomerase activity through direct phosphorylation of the hTERT protein, the catalytic component of the telomerase complex. Additionally, Akt activation leads to mTOR phosphorylation, a kinase that regulates cell growth and proliferation by integrating signals from growth factors, hormones, nutrients and energy status [18].

### 2.3. mTOR

mTOR is a member of the phosphatidylinositol 3-kinase-like protein kinase (PIKK) family and has a C-terminal catalytic domain with sequence homology to PI3Ks. mTOR consists of a multi-domain protein with protein kinase activity, adding phosphate groups to serine or threonine residues, while it participates in a wide range of molecular pathways, many of which are involved in anabolic metabolism [19]. The main regulators of mTOR are insulin and insulin growth factors (IGFs), amino acids, various forms of stress and the accessibility of ATP [20]. mTOR is a basic component of the two different mTOR complexes: mTOR complex 1 (mTORC1) and mTOR complex 2 (mTORC2). mTORC1 activation results in phosphorylation of the downstream effectors p70S6K and 4EBP1 to regulate cell metabolism via mRNA translocation and protein synthesis. On the other hand, the mTORC2 complex promotes the cellular actin cytoskeleton via phosphorylation of Akt at ser473 [21,22]. As multiple genes of the PI3K/Akt/mTOR signaling pathway are frequently found mutated in cancer and interact with other crucial genes in oncogenesis, such as *PTEN*, they arguably constitute important therapeutic targets in the cancer therapeutic field [23,24].

## 3. The PI3K/Akt/mTOR Signaling Pathway in Human Leukemias and Lymphomas

PI3K/Akt/mTOR pathway activation in lymphomas is associated with over-expression of various cytokines and growth factors, such as IL-6, IL-10 and the platelet-derived growth factor (PDGF). The relationship between the expression of activated Akt and the tumor suppressor gene *PTEN* in lymphomas is well established [25].

Across patients with B cell malignancies, *PIK3CA* mutations or amplification were found in 8% of DLBCL cases, mainly in the catalytic domain, and in 68% of MCL cases, respectively [26,27]. Amplification of *PIK3CA* in CLL patients has been reported in 5.6% of patients [28]. In a study of bone marrow trephines of 77 HCL patients, our group reported that most (74/77) cases expressed phosphorylated (p)Akt but not p-mTOR (10/77 cases) [3].

*PTEN* abnormalities have been found in hematologic malignancies in lower frequencies, as compared to that of solid tumors [29]. The highest rates of *PTEN* deletion or mutation have been found in acute T cell lymphoblastic leukemia (T-ALL), while few *PTEN* alterations have been reported in acute myeloid leukemia (AML) and acute B cell lymphoblastic leukemia (B-ALL). Deletions or mutations of *PTEN* are also found in T cell lymphomas and large B cell lymphomas, respectively. High *PTEN* mutation frequency in T-ALL and T cell lymphomas is probably related to the essential role of the *PTEN* or *PI3K* pathways in T cell lineage commitment and differentiation, while the B cell is regulated by an intermediate level of PI3K signaling. Across B cell lymphoma patients, PTEN loss is observed in 15% of MCL cases, in 37–55% of DLBCL cases and in 21% of follicular lymphoma (FL) cases [27,30,31,32,33]. As far as DLBCL is concerned, *PTEN* loss was found in 55% of germinal center B cell type (GCB) DLBCL patients, as opposed to only 14% of non-GCB DLBCL ones [31]. In GCB DLBCL, *PTEN* deletions and amplification of the microRNA-17-92 cluster (MIR17HG) sustain cell proliferation [34], while similar deregulations are also in place in Burkitt lymphoma (BL) cells, in which the constitute activation of the *MYC* oncogene directly activates the PI3K/Akt/mTOR pathway [35,36].

### 3.1. Diffuse Large B Cell Lymphoma

DLBCL represents the most common subtype of non-Hodgkin lymphoma and accounts for approximately 30% of newly diagnosed cases. Recent studies have demonstrated the aberrant expression of the PI3K/Akt/mTOR signaling pathway and its crucial role in controlling cell proliferation and survival in lymphoid malignancies [37,38]. More specifically, Uddin et al. showed that the constitutive activation of the PI3K/Akt pathway is constitutively activated in human DLBCL cell lines [37]. They also highlighted that in most DLBCL cells, the inhibition of PI3K leads to apoptosis through the release of mitochondrial cytochrome C and the activation of downstream caspases, while patients with high p-Akt expression showed poor survival [37]. Additionally, Fillmore et al. observed that DLBCL lymphomas are frequently characterized by a higher level of p-Akt, providing additional support for its positive role in DLBCL cell growth [39]. The activation of the PI3K/Akt/mTOR pathway in DLBCL results in PTEN loss, additional mutations or constitutive activation of upstream regulatory pathways. PTEN is overexpressed in the majority of the ABC subtypes of DLBCL, while the GCB subtype is defined by the loss of PTEN protein expression [40]. In the GCB subtype, the activation of the PI3K/Akt/mTOR pathway and the consequent loss of PTEN correlate with the phosphorylation of Akt. In contrast, in ABC DLBCL, this specific activation is not associated with PTEN loss, which implies that there may be a different mechanism underlying PI3K/Akt/mTOR pathway activation in each DLBCL subtype [31]. The constitutive activation of nuclear factor κB (NF-κB) due to activation of the “CBM” signaling complex consisting of CARD11, BCL10 and MALT1 represents the hallmark of the ABC subgroup of DLBCL [41,42]. The “CBM” signaling complex may be constitutively stimulated by point mutations of *CARD11* and through chronic active B cell receptor (BCR) signaling and downstream kinases, including PI3K [41,42]. In fact, the phosphorylation of CD19 by BCR leads to the binding of the regulatory subunit p85 and the recruitment of the p110 catalytic subunit [43]. A downstream protein of BCR, BTK is activated by PI3K (through PIP3) in B cells. Recurrent mutations in *PIK3CD, PIK3R1* and *mTOR* occur in DLBCL and concern different molecular targets with *mTOR* mutations found in the HEAT domain (A835T) [44,45]. In the GCB DLBCL cell lines, an antigen-independent BCR signaling activates Akt, regulating the proliferation and the size of the tumor, and Akt knockout results in rapid cell reduction [46]. In ABC DLBCL, PI3K regulates the activation of NF-κB, which represents a typical feature of this DLBCL subgroup [47].

### 3.2. Mantle Cell Lymphoma

MCL represents 6% of all non-Hodgkin lymphomas and, until recently, had a median survival of 4 years, which has been increased with the use of novel agents and cellular therapies [48]. MCL is associated with the translocation t(11;14)(q13;32), with consequent cyclinD1 overexpression [49]. Among the signaling pathways that may be deregulated in MCL cells, the PI3K/Akt/mTOR pathway has recently attracted great interest as a potential therapeutic target as described in detail in the following sections. Specifically, the PI3K/Akt/mTOR pathway regulates the expression of the cell cycle protein p27 and, via the phosphorylation of threonine 157, finally prevents its functions [50]. Moreover, Akt, through an inhibition of kinase GSK-3 that negatively regulates cyclinD1 expression, regulates the levels of cyclinD1 [51]. Recent gene profiling studies have demonstrated that the PI3K/Akt/mTOR pathway is upregulated in MCL and evidence has also been provided suggesting that Akt and mTOR-dependent signaling may be constitutively active in MCL [52,53,54].

## 4. Clinical Applications of m-TOR Inhibitors in B Cell Lymphomas

The mTOR inhibitors have been used in several clinical trials for almost all different histological types of B cell malignancies, either as monotherapy or in combination with other regimens (chemotherapy, immunotherapy, chemoimmunotherapy, etc.). Temsirolimus, an intravenously (iv) administered mTOR inhibitor, was commercially approved for relapsed or refractory (R/R) MCL in 2009. Everolimus is an oral mTOR inhibitor, which has been evaluated in B cell lymphomas but has not been approved by the US or European regulatory authorities. Everolimus has also shown clinical activity in classical Hodgkin lymphoma with responses seen even in heavily pretreated patients, but its development was overcome by the appearance of anti-CD30 and anti-PD-1 targeting with brentuximab vedotin and the checkpoint inhibitors nivolumab and pembrolizumab [55,56,57]. Thus, this review will focus to the role of mTOR inhibitors in B cell non-Hodgkin lymphomas with an emphasis on the role of temsirolimus in MCL.

### 4.1. Clinical Trials with mTOR Inhibitors in Mantle Cell Lymphoma

#### 4.1.1. Temsirolimus

##### Temsirolimus Monotherapy

The initial phase II studies of temsirolimus monotherapy, published in 2005 [58] and 2008 [59], provided similar and very promising results using highly diverse dosages of 250 mg [58] or 25 mg per week [59] with a fixed treatment duration not exceeding 12 months, as summarized in Table 1. In the first one [58], 35 heavily pretreated patients with R/R MCL, who had received a median of three prior treatment lines (range 1–11), received temsirolimus at a dose of 250 mg iv weekly. The investigators observed a 38% overall response rate (ORR), with only 3% complete remission (CR). The main toxicity was hematologic of grade 3 or higher in the majority (71%) of patients. All except four patients required dose reduction, but generally hematologic toxicity was reversible after the predefined dose adjustments. Therefore, the North Central Cancer Treatment Group conducted an additional phase II trial [59] with temsirolimus monotherapy using a low-dose schema of 25 mg iv every week in 28 patients with R/R MCL who had received a median of four prior treatment lines (range 1–9). The ORR was 41% but again with a very low CR rate of 4%. In this study, hematologic toxicities were less frequent, i.e., grade 3 or higher in 50% of the patients. In both trials, the median time to progression and the median duration of response were remarkably similar, ranging between 6 and 7 months (Table 1).

A registrational trial [60] was conducted following these promising results, in which 162 patients with R/R MCL were randomly assigned in an 1:1:1 fashion to receive either one of two temsirolimus dosing schedules (175 mg weekly for three weeks followed by either 75 mg (175/75-mg) or 25 mg (175/25-mg) weekly) or the investigator’s choice. In contrast to the previous phase II trials, temsirolimus was continued until progressive disease (PD) or unacceptable toxicity. The 25 mg dose level was selected based on the favorable results of the 2008 phase II trial reporting similar efficacy with lower toxicity [59], while the 75 mg dose level was selected because it yielded higher blood concentrations than the 25 mg schedule permitting the evaluation of a dose–response effect. The investigator’s choice therapies included any monotherapy among gemcitabine, fludarabine, chlorambucil, cladribine, etoposide, cyclophosphamide, thalidomide, vinblastine, alemtuzumab or lenalidomide. The patients were again heavily pretreated, having received a median of three and four prior lines of therapy in the temsirolimus and investigator’s choice arms, respectively (range 2–7). The median time for response to temsirolimus was ~3.5 months for both groups and the ORR was 22% (2% CR) for the 175/75-mg arm versus only 6% (no CRs) for the 175/25-mg arm. The median progression free survival (PFS) was 4.8, 3.4 and 1.9 months for the temsirolimus 175/75-mg, 175/25-mg and the investigator’s choice groups, respectively, being significantly longer with the 175/75 mg schedule than with the investigator’s choice (*p* = 0.0009; HR (HR) 0.44). However, this was not translated into any significant overall survival (OS) difference (median 12.8 months and 9.7 months, *p* = 0.35). This improvement in disease control came with an acceptable toxicity profile. The most frequent grade 3/4 adverse events in the temsirolimus groups were thrombocytopenia, anemia, neutropenia, and asthenia, but all were manageable with dose reductions or increasing or dose intervals. Based on these results, temsirolimus gained approval for R/R MCL at the 175/75 mg weekly dose schedule.

The significance of the 3 weekly 175 mg doses remains, however, obscure. Thus, Jurczak et al. conducted a phase IV randomized trial comparing the approved 175/75 mg weekly dose schedule with a fixed 75 mg weekly dose from the beginning [61]. Temsirolimus was again given until PD or unacceptable toxicity. In total, 47 and 43 patients who were heavily pretreated with R/R MCL received the above dose schedules with similar ORR rates (28% and 21%, respectively) and minimal CR rates (4% and 2%). PFS was also similar (Table 1) as was the duration of response, and all these figures were very similar to those of the registrational trial published by Hess et al. [60]. When PFS was assessed by the investigators, the median PFS was significantly longer in the approved dose schedule (4.7 versus 3.9 months, HR 0.65 with 95% CI 0.45–0.92). Serious adverse events were also lower in the 175/75 mg arm. Based on these results, the authors concluded that the approved 175/75 mg schedule should remain the preferred dosing regimen. At the end, only 2/90 patients remained on treatment and progression-free beyond 3 years.

Overall, the above trials have provided similar results which suggest that temsirolimus can induce mainly partial responses in 20–40% of patients with R/R MCL, and a small minority may achieve durable responses lasting >3 years. Thus, it may provide a temporary treatment option in order to enable the implementation of a more effective or radical therapy. However, these results have been produced in the past in patient populations that had not been exposed to Bruton kinase inhibitors, such as ibrutinib and probably not to lenalidomide. The rapid progress in the treatment of R/R MCL made the comparison of temsirolimus with ibrutinib necessary. Thus, the phase III RAY trial [62] enrolled 280 R/R MCL patients who were randomized either to receive daily oral ibrutinib 560 mg or iv temsirolimus at the approved 175/75 mg schedule until PD or unacceptable toxicity. This cohort was less heavily pretreated with a median of two prior lines of therapy (range 1–9). There was a highly significant difference in the ORR (72% versus 40%, *p* < 0.001) and the CR rate (19% versus 1%) (Table 1). The median PFS for ibrutinib was 15.6 vs. 6.2 months for temsirolimus (HR 0.45, 95% CI 0.35–0.60, *p* < 0.0001) [63]. The superiority of ibrutinib extended to all patient subgroups except the small subgroup of patients with the blastoid variant. Interestingly, although ibrutinib was much more effective in the second-line setting, the outcomes after temsirolimus were similar in patients with either 1 or >1 prior lines of therapy. Ibrutinib was less toxic than temsirolimus, with grade 3 or higher AEs reported in 68% versus 87% of the patients. As a result, only 6% of the patients discontinued ibrutinib vs. 26% from the temsirolimus arm due to toxicity.

Thus, although effective in R/R MCL, temsirolimus was overcome by the development of ibrutinib [64], other BTK inhibitors and other recent options, such as CAR-T cell therapy with brexucabtagene autoleucel [65].

**Table 1 ijms-24-14110-t001:** Summary of phase 2 and phase 3 trials of temsirolimus either as monotherapy or in combination with rituximab in relapsed/refractory MCL.

	Witzig T. et al., 2005 [58]	Ansell S. et al., 2008 [59]	Hess G. et al., 2009 [60]	Hess G. et al., 2009 [60]	Ansell S. et al., 2011 [66]	Dreyling M. et al., 2016 [62]		Jurczak W. et al., 2018 [61]	Jurczak W. et al., 2017 [61]
**Setting**	Phase 2	Phase 2	Phase 3	Phase 3	Phase 2	Phase 3		Phase 4	Phase 4
**Patients**	35	28	54	54	69	141		47	43
**Dosing schedule**	250 mg/wk	25 mg/wk	175 mg/wk × 3 wks then 75 mg/wk	175 mg/wk × 3 wks then 25 mg/wk	25 mg/wk Rituximab	175 mg/wk × 3 wks then 75 mg/wk		175 mg/wk × 3 wks then 75 mg/wk	75 mg/wk
**Treatment duration**	CR + 2 cycles, if CR at 6 mo 12 cycles, if PR at 6 mo 6 cycles, if SD at 6 mo or until PD at anytime	CR + 2 cycles, if CR at 6 mo 12 cycles, if PR at 6 mo 6 cycles, if SD at 6 mo or until PD at anytime	until PD or UT	until PD or UT	CR + 2 cycles, if CR at 6 mo	until PD or UT		until PD or UT	until PD or UT
**Age (median) (range)**	70 (30–89)	69 (51–85)	68 (44–87)	68.5 (43–85)	67 (44–86)	68 (13)		66 (47–85)	67 (47–86)
**Gender (male, %)**	74	68	85	74	72	77		72	84
**Previous therapies (median) (range)**	3 (1–11)	4 (1–9)	3 (2–7)	3 (2–7)	2 (1–9)	2 (1–9)		3 (2–7)	2 or 3 (1–5)
**Blastoid (%)**	NA	14	0	17	NA	12		15	16
**PS ≥ 2 (%)**	12	22	19	15	4	1		11	16
**PS range**	0–2	0–2	60–100	60–100	0–2	0–2		0–2	0–2
**ORR (%)**	38	41	22	6	59	40		28	21
**CR (%)**	3	4	2	0	19	1		4	2
**PFS (median)**	6.5 mo	6 mo	4.8 mo	3.7 mo	9.7 mo	6.2 mo		4.3 mo	4.5 mo
**DOR (median)**	6.9 mo	6 mo	7.1 mo	3.6 mo	10.6	11.0 R-sens	7.0	9.0 mo	8.7 mo
						6.0 R-ref			
**Refractory**	54	50	NA	NA	NA	33		NA	NA

CR: complete remission, DOR: duration of response, MCL: mantle cell lymphoma, mo: months, NA: not applicable, ORR: overall response rate, PD: disease progression, PFS: progression free survival, PR: partial remission, PS: performance status, SD: stable disease, UT: unacceptable toxicity, wk: week.

##### Temsirolimus Combinations

In the same population of R/R MCL patients, temsirolimus was also tested in a phase II study [66] at a dose of 25 mg/week in combination with rituximab for a maximum of 12 cycles of 28-day duration (Table 1). The trial enrolled 69 patients who had received a median of two prior lines of therapy (range 1–9). Compared to temsirolimus monotherapy, the results with this fixed-duration, low-dose regimen were very satisfactory, with an ORR of 59% including 19% CRs. The combination of R–temsirolimus was effective in the subgroup of rituximab-refractory patients with an ORR of 52% versus 63% in rituximab-sensitive patients. The median PFS was 9.7 months, and the median duration of response was 10.6 months (11.0 versus 6.0 for rituximab-sensitive and refractory patients, respectively).

Temsirolimus has also been tested in more complex combinations with immunochemotherapy in MCL, either in the R/R setting or even in the first-line, as summarized in Table 2.

In a phase IB study of 41 patients with R/R MCL from the LYSA working group [67], temsirolimus was tested in several doses ranging from 15 mg to 75 mg along with R-CHOP (*n* = 10), R-FC (*n* = 14), or R-DHA (*n* = 17). The maximum tolerated dose of temsirolimus was 15 mg for the R-CHOP-T arm but was not determinable for the other regimens due to toxicity, as even stepping down to the 15 mg dose was associated with dose-limiting adverse events in the R-FC-T and R-DHA-T arms. The ORR to the three regimens ranged from 40% to 47%. Interestingly, six patients of the R-DHA-T arm (35%) reached CR, while the CR rates for the other two arms was 20% and 21%. The median PFS ranged from 8.6 to 15.1 months (Table 2). All patients experienced at least grade 3 adverse events, mainly thrombocytopenia (76%). Overall, the temsirolimus–immunochemotherapy combinations evaluated in this study were limited by rather unacceptable toxicity with modest efficacy (Table 2).

In another phase I/II study of 29 R/R MCL and 10 R/R FL patients [68], temsirolimus was given in doses ranging from 25 to 75 mg in phase I and 50 mg in phase II, on days 1, 8 and 15 of 28-day cycles in combination with bendamustine and rituximab (BeRT) for a total of four cycles. Among the 39 patients with MCL or FL, 9 (23%) had previously received bendamustine. The median number of prior regimens was two (range 1–3). Objective response (best response) was observed in 33/39 patients (89%; 24 MCL (89%) and 9 FL (90%)), including 14 CRs (38%; 12 MCL (36%) and 2 FL (20%)). The median PFS was estimated at 1.5 years for MCL and 1.82 years for FL, and the median OS had not been reached for either lymphoma subtype. Grade 3/4 adverse events (AEs) were mainly hematologic, while the non-hematologic grade 3 and 4 AEs were very infrequent. These results appear promising for the R/R setting with a median of two prior regimens that were achieved with acceptable toxicity. Thus, they may deserve further evaluation in larger patient series with variable treatment histories.

The combination of temsirolimus with bortezomib has been applied in a small series of seven patients with R/R MCL with an ORR of 57% in the context of a trial including a broader spectrum of B cell NHLs as described below in Section 4.2.1 [69].

Finally, five planned iv temsirolimus dosage schedules were combined with fixed doses of rituximab and cladribine in a phase I study of 17 previously untreated patients with MCL who were not scheduled for autologous stem cell transplantation [70]. The five planned temsirolimus i.v. doses were 15 mg day 1, 25 mg day 1, 25 mg days 1 + 15, 25 mg days 1 + 8 + 15 and 25 mg days 1 + 8 + 15 + 22. The ORR was 94% with 53% CR and 41% PR. Most patients (59%) had an intermediate-risk MCL International Prognostic Index (MIPI) and 35% had high-risk MIPI. The median PFS was 18.7 months; at 3 years, the PFS rate was just above 30%. It is possible that higher temsirolimus doses might have been tolerated with this regimen, as there were no dose-limiting events in this study. However, despite the high ORR, PFS appears to be suboptimal for previously untreated MCL.

**Table 2 ijms-24-14110-t002:** Studies combining Temsirolimus and immunochemotherapy in R/R or previously untreated MCL.

	LYSA Working Group [67]	Hess et al. [60]	Inwards et al. [70]
**Setting**	Phase IB, dose-escalation study	Phase I/II	Phase I dose-escalation study
**Patients**	41	29	17
**Dosing schedule**	R-CHOP, R-FC, RDHA and T (15 to 50 mg weekly)	BR and T (15–75 mg in phase I and 50 mg in phase II on D1,8, 15 of 28-day cycles)	R-Cladribine and T (15 mg D1 to 25 mg in 4 dose levels (1–4 weekly applications per cycle))
**Treatment duration**	4 cycles plus 2 in case of clinical benefit	4 cycles	28-day cycles, up to 4 or 6 depending on response
**Age (median (range))**	68 (56–79)	73 (46–79)	75 (52–86)
**Gender (male, %)**	78	72	65
**Previous therapies, (median (range))**	1 (1–3)	2 (1–3)	0 (first-line)
**Blastoid (%)**	7	NR	NR
**PS range**	0–2 *	0–2 **	NR (0–3 allowed)
**Temsirolimus MTD or RP2D**	15 mg	50	MTD not reached, RP2D NA (phase II part abandoned)
**ORR (%)**	R-CHOP-T: 40 RFC-T: 43 R-DHA-T: 47	89	94
**CR (%)**	RCHOP-T: 20 RFC-T: 21 R-DHA-T: 35	44	43
**PFS (months)**	R-CHOP-T: 15.1 RFC-T: 8.6 R-DHA-T: 13.9	18	18.7
**DOR (months)**	R-CHOP-T: 7.9 RFC-T: 13.5 R-DHA-T: 16	NR	NR
**OS (months)**	R-CHOP-T: NR RFC-T: 10.1 R-DHA-T: 24.2	56% at 3 years	NR

BR: bendamustine-rituximab, CR: complete remission, DOR: duration of response, LYSA: lymphoma study association, MCL: mantle cell lymphoma, MTD: maximum tolerated dose, NA: not applicable, NR: not reported, ORR: overall response rate, OS: overall survival, PFS: progression-free survival, PS: performance status, R-CHOP: rituximab-cyclophosphamide-doxorubicin-prednisone, R-DHA: rituximab-high-dose cytarabine, RFC: rituximab-fludarabine-cyclophosphamide, RP2D: recommended phase II dose, R/R: relapsed/refractory, T: temsirolimus. * PS ≥ 2 in 12.2%, ** no patient with PS 2.

#### 4.1.2. Everolimus

The efficacy of another available mTOR inhibitor, everolimus, in R/R MCL patients, has been tested in several trials but was found less promising. In a phase II study from the Mayo Clinic [71], 77 patients with various histologic subtypes of aggressive B cell NHLs—among them 19 with R/R MCL—received oral everolimus at the conventional dose of 10 mg/d until PD. The ORR for the latter subgroup was 32% (6/19), with two patients in CR and four in PR.

Two subsequent studies of everolimus monotherapy focused specifically on R/R MCL patients. The first one, a phase II study [72], was conducted by the Swiss SAKK and the French GOELAMS groups from the European Mantle Cell Lymphoma Network and included 36 R/R MCL patients. Patients should have received a maximum of three prior treatment lines and were treated with 10 mg/d oral everolimus for a total of six 28-day cycles or until PD. The response rate among the 35 evaluable patients was only 20%, with two CR and five PR. The median PFS was 5.5 months for all patients and 17 months for those who managed to receive >6 cycles. However, the toxicity profile was favorable, with the most common grade 3 or higher AEs being hematologic and occurring in less than 10% of patients.

Subsequently, in PILLAR-1, the second phase II study [73], 58 R/R MCL patients who were considered refractory or intolerant to bortezomib received everolimus 10 mg/d as monotherapy. Among them, 85% were refractory to bortezomib and had received a median of thee prior regimens. The primary endpoint was ORR, with the targeted ORR set at 20%; an ORR ≤ 7% would have precluded further investigation. If at least 3 out of 34 patients enrolled in part 1, 23 additional patients would have been evaluated, and everolimus would have been worthy of consideration for clinical interest in this clinical setting if 8/57 patients (14%) experienced a response. The study did not meet its primary endpoint, as only five PRs (8.6%) were observed. The median PFS was 4.4 months, and the median OS was 16.9 months.

### 4.2. Clinical Trials with m-TOR Inhibitors in Other (Non-MCL) of B Cell Malignancies

#### 4.2.1. Temsirolimus

In contrast to MCL, the efficacy of mTOR inhibitors and especially temsirolimus in non-MCL B cell malignancies is less promising. In a phase II trial, 89 patients with R/R non-MCL B cell malignancies received 25 mg weekly temsirolimus iv [74]. Among them, 32 had DLBCL and transformed FL (group A), 39 FL (group B) and 18 others had indolent lymphomas (mainly CLL/small lymphocytic lymphoma (SLL); group C). Group A had an ORR and s CR rate of 28.1% and 12.5%, respectively, with a median PFS of 2.6 months and a median OS of 7.2 months. Group B had an ORR and a CR rate of 53.8% and 25.6%, respectively, and a median PFS of 12.7 months; the median OS has not yet been reached. In Group C, only 11% PRs were observed with no CR. Although toxicity generally consisted of mild myelosuppression and mucositis, pneumonitis was also observed in four patients.

Because the proteasome inhibitor, bortezomib, had shown efficacy as monotherapy [75,76,77,78] for R/R MCL and other B cell malignancies, the Wisconsin Oncology Network tested the combination of temsirolimus with bortezomib in a phase II study [69] with 39 R/R B-cell NHLs. The majority were DLBCL. In the whole cohort, the CR and PR rates were 7.7% (three patients) and 23% (nine patients). The responses according to histologic subtype were the following: DLBCL, 3 out of 18 patients (16.7%); MCL, 4 out of 7 patients (57%); and FL, 5 out of 9 patients (56%). There were no responses among the patients who had SLL or marginal zone lymphoma. The median PFS was 4.7 months; it was only 2 months for DLBCL but 7.5 months for MCL and 16.5 for patients with FL. The most common grade 3 or higher AEs were hematologic, while common grade 1 and 2 toxicities were rash, hyperlipidemia, diarrhea, constipation and peripheral neuropathy.

Finally, in a phase 2 trial, 37 patients with R/R primary central nervous system (PCNS) lymphoma received temsirolimus monotherapy at a dose of 75 mg iv weekly (25 mg in the cohort of the six initial patients) [79]. These patients had failed methotrexate-based regimens and were ineligible or had failed ASCT. The median age of the patients was 70 years (range 22–83) and the median performance status (PS) was two (range 0–2). Impressively, temsirolimus monotherapy resulted in remission in 20 patients (54%), with 21.8% (8 patients) achieving CR or CR unconfirmed (CRu). Although these responses were promising, they were short-lived with a median PFS of 2.1 months. Interestingly, the temsirolimus cerebrospinal fluid (CSF) concentration was 2 ng/mL in one patient in the 75 mg cohort; in all others, temsirolimus was not detectable in CSF. This apparent discrepancy between the efficacy and the absence of temsirolimus or its metabolites in the CSF is not readily understood and potential reasons have been speculated in the discussion of the publication (for example CSF levels not reflecting brain parenchyma concentrations, breakdown of the normal blood–brain barrier within the tumor bulk, etc.). Toxicity reflected the poor condition of patients with PCNS lymphoma: there were 28 severe adverse events in 21 patients and 10 patients died within 4 weeks from the last treatment administration; 5 from disease progression and 5 (13.5%) from toxicity (pneumonia 2, sepsis 2, cerebral hemorrhage 1).

#### 4.2.2. Everolimus

Although everolimus, has been more extensively studied in non-MCL B cell malignancies, its clinical activity proved modest as well.

##### Everolimus for R/R Aggressive B Cell Lymphomas

Everolimus has been tested in R/R aggressive B cell lymphomas (mainly DLBCL) in several trials either as monotherapy or in combination with rituximab, lenalidomide, bendamustine or panobinostat.

In the previously mentioned phase II study from the Mayo Clinic with 77 R/R aggressive lymphoma cases [71], there were 47 patients with DLBCL who received 10 mg/d oral everolimus monotherapy and achieved an ORR of 30% but no CRs (14 patients, all in PR).

In another phase II study, 26 R/R DLBCL patients received the combination of everolimus 10 mg/d and Rituximab for a maximum of 12 cycles or until PD [80]. The median number of prior treatment lines was four. Among the 24 patients evaluable for response, there were two CRs and six PRs, resulting in an ORR of 38%. Eleven patients had to reduce or temporarily discontinue their treatment due to toxicity, but the majority (18 patients) discontinued due to PD.

In a Mayo Clinic phase I/II study [81], 55 patients with various R/R B cell malignancies or Hodgkin lymphoma were recruited. Patients received everolimus in combination with lenalidomide in four different dose schedules. The maximum tolerated dose (MTD) was estimated to be the combination of 5 mg/d everolimus and 10 mg/d lenalidomide. Lenalidomide was given for 21 days of each 28-day cycle. Among the 23 DLBCL cases, there were only five PRs (22%).

In another phase II study, everolimus and/or panobinostat were evaluated in a population of 33 patients with heavily pretreated R/R DLBCL, as the median number of prior regimens was 2–4 and most were refractory to the last therapy. In a rather complex design, the patients could receive either the two agents sequentially for up to six cycles each and then the combination or the combination from the beginning. The results were disappointing: only 1/9 patients initially treated with everolimus responded by achieving CR, which lasted for 3 months off-therapy. No responses were seen with panobinostat and none of the patients enrolled in the sequential part achieved receiving the combination. Among 16 evaluable patients, 4 achieved a PR with no CRs (25%), but all four responses were short-lived. The final selected dose scheme was panobinostat i.v. 15 mg t.i.w. and everolimus p.o. 7.5 mg/d. Despite amendments with dose reductions, all patients enrolled in the doublet therapy experienced grade 3/4 AEs and half of them serious AEs [82].

Finally, in a recent phase I study of 18 patients with R/R Hodgkin and NHL [83], everolimus was tested with the combination of bendamustine and rituximab. The MTD for everolimus in this study was 7.5 mg/d. Only one of the five DLBCL patients (20%) responded with PR.

Overall, the results of everolimus monotherapy or combinations in R/R DLBCL point out to modest activity; hence, everolimus was not further developed in this indication.

##### Everolimus in the First-Line Treatment of Aggressive B Cell Lymphomas

Everolimus has also been studied in the frontline setting for DLBCL patients. In a Phase I study from the Alliance (NCCTG N1085) [84], 24 untreated DLBCL patients received R-CHOP-21 with oral everolimus at various dose schedules with pegfilgrastim support given on day 2 of each cycle. The MTD for everolimus was estimated at 10 mg/d for days 1–14. Twenty-three out of 24 patients achieved CR, as assessed by positron emission tomography combined with computed tomography (PET-CT). The last patient withdrew from the study but achieved CR with R-CHOP only. In a more recent update of this cohort with a median follow-up time of 37.2 months [85], the 24-month event-free survival (EFS) and OS were 91.7% (22/23) and 100%, respectively. The median time from diagnosis to treatment in this series was 14 days. These observations compared favorably with various previous reports of R-CHOP for DLBCL in a frontline setting and among them with data from the same study group, which had shown a 24-month failure rate of 44% for those receiving R-CHOP within less than 15 days from diagnosis and 28% for those who received treatment later [86]. Notably, a 24-month EFS is a useful surrogate marker for long-term remission and the rate of 91.7% observed in this trial is outstanding. Despite these promising results, the R–Eve–CHOP combination was not selected for further clinical development.

##### Everolimus as Adjuvant Therapy in DLBCL

Interestingly, the most important scientific evaluation of everolimus in DLBCL was performed in the setting of maintenance therapy after a complete metabolic response to immunochemotherapy in patients with a high-risk International Prognostic Index (IPI), defined as IPI ≥ 3. In the PILLAR-2 trial, 742 DLBCL patients in first CR after R-CHOP were randomized to receive either 10 mg/d oral everolimus for 1 year or a placebo [87]. Everolimus at a dose of 10 mg/d showed no improvement over placebo, with increased toxicity as expected. The 2-year DFS were 77.8% vs. 77.0%, respectively (HR 0.92; 95% confidence intervals 0.69–1.22, *p* = 0.28). Similarly, the 2-year OS rates were 90.7% vs. 88.3%, respectively (HR 0.75; 95% confidence intervals 0.52–1.09). With 7% versus 12% lymphoma-specific survival (LSS) events reported in each group, the HR was 0.66 (95% confidence intervals 0.41–1.07), which is marginally but not significantly better in favor of everolimus. In an exploratory subgroup analysis, males, younger patients (<65 years old), those of Asian ancestry and those with IPI 4–5 had marginally or even significantly better DFS and OS. The results of PILLAR-2 are summarized in Table 3.

Grade 3 or 4 AEs were reported in 54% of patients in the everolimus arm and 25% of patients in the placebo arm. The three most reported AEs were stomatitis, diarrhea and neutropenia with everolimus and neutropenia, fatigue, and diarrhea with the placebo. The discontinuation rates for AEs were 30% versus 12% in the everolimus and the placebo arms, respectively. The discontinuation rates for progressive disease were 6% versus 13%. Thus, the actual dose intensity for everolimus was 7.9 mg/d versus 9.4 mg/d for the placebo, and the median durations of the treatments were 48.4 versus 52.0 weeks.

##### Everolimus for Indolent B Cell Lymphomas

In the subgroup of indolent B cell malignancies, everolimus has shown very modest activity with a problematic toxicity profile. The efficacy results appear more promising only for the subgroup of FL patients.

The International Extranodal Lymphoma Study Group conducted a phase II study recruiting 30 patients with R/R marginal zone lymphomas (MZL) [88]. Twenty had extranodal, six splenic and four nodal MZLs and were treated with everolimus monotherapy at a dose of 10 mg/d of 28-day cycles until PD or unacceptable toxicity. The latter occurred in 17 patients. The ORR of the evaluable patients was 25% (one CR, five PR). The median PFS was 14 months, and the duration of response (DoR) was 6.8 months. Grade 3/4 AEs were common: neutropenia and thrombocytopenia (17% of patients each), infections (17%), mucositis and dental infections (13%) and lung toxicity (3%).

In another phase II study of 55 patients with R/R indolent B cell lymphoma, mainly CLL/SLL patients (n = 26) and FL (n = 23) [89], everolimus was given at 10 mg/d in a 28-day cycle. The median number of prior treatment lines was five (1–10). The ORR was 61% (14/23) in patients with FL and 19% (5/26) in CLL/SLL. The median DoR was 11.5 months (range, 5.7–30.4), and the median PFS was 7.2 months (range, 5.5–12.5). Thirty-five percent of the patients (19/55) experienced non-hematologic toxicity (infection, metabolic alterations, mucositis, rash, etc.). In the subgroup of Waldenstrom macroglobulinemia (MW), everolimus has shown some activity but with increased toxicity.

In a phase II study of 50 patients with R/R MW30 who received everolimus at 10 mg/d, the ORR was 70%, with 42% PR and 28% minimal response [90]. The estimated 12-month PFS was 62%. Grade 3 or higher AEs were observed in 56% of patients. The most common were hematologic. Pulmonary toxicity occurred in 10% of patients. Fifty-two percent of patients had a reduced everolimus dose due to toxicity.

## 5. Conclusions

The PI3K/Akt/mTOR signaling pathway plays a critical role in a variety of cellular processes, such as cell proliferation and survival, and in lymphomas it is associated with the over-expression of various cytokines and growth factors in malignant lymphomas.

The mTOR inhibitors, temsirolimus and everolimus, have been tested in a range of clinical trials enrolling patients with B cell malignancies, either as a monotherapy or in combinations. Most of them are small phase I and II trials, while only three phase III trials have been conducted [60,62,87].

The B cell malignancy with the most promising results when treated with mTOR inhibitors is MCL. As a result, temsirolimus has been granted commercial approval for R/R MCL since 2009. Temsirolimus remains a valid option for R/R MCL, but its efficacy is modest and superseded by the new era of BTK inhibitors and CAR-T cells [91].

No other commercial authorization exists for mTOR inhibitors in B cell malignancies. For patients with other indolent B cell malignancies, temsirolimus and everolimus have shown moderate activity [69,74,88,89,90] except for the FL subgroup [69,74,89]. FL cases respond more favorably but with the trade-off of toxicity. The B cell malignancy with the worst clinical outcome with mTOR inhibitors is SLL/CLL [74,89].

For R/R aggressive B cell lymphomas, the efficacy of mTOR inhibitors was very modest with increased toxicity [69,74,79,80,81,83]. In the front-line setting, everolimus can be safely and effectively combined with R-CHOP [85], but no large phase III confirmatory studies have been conducted. The R–Eve–CHOP combination was not further developed. Meanwhile, other potential combinations either gained approval (polatuzumab vedotin) or were intensively evaluated in phase III studies (epcoritamab, tafasitamab plus lenalidomide, etc.) in combination with R-CHOP with or without vincristine [92]. The adjuvant use of everolimus as maintenance after chemoimmunotherapy was not beneficial overall in the PILLAR-2 study [87]. Potential benefits in specific subgroups in exploratory subgroup analysis may truly exist but were not sufficient for the approval of everolimus in this setting.

As MCL and other indolent lymphomas remain incurable, mTOR inhibitors, especially temsirolimus, might be further tested in combination with other agents in the future. A potential place for off-label mTOR inhibition by everolimus could be relapsed/refractory classical Hodgkin lymphoma, where novel agents are warranted for patients failing PD-1 inhibitors either as monotherapy or in combinations or in the context of metronomic therapy [93]. Further to the above, preclinical data suggest that combined PI3Kβ/δ and mTOR targeting with AZD8186 and AZD2014, respectively, may overcome resistance to PI3Kβ/δ inhibition and completely prevent outgrowth of lymphoma cells in vivo in cell line- and patient-derived xenograft mouse models of DLBCL. This appears applicable in both germinal center cell and activated B cell-derived DLBCL and may justify the evaluation of a combined PI3Kβ/δ and mTOR inhibition in clinical trials [94]. Overall, mTOR inhibition is an existing option in MCL, may be used off-label in some other clinical settings and may also provide a future partner for new effective combinations with chemotherapy or other targeted therapies.

## Figures and Tables

**Table 3 ijms-24-14110-t003:** Summary of efficacy and toxicity results in the PILLAR-2 study, comparing everolimus with placebo as maintenance therapy in high-risk DLBCL patients achieving complete metabolic remission with immunochemotherapy.

	Everolimus Arm (n = 372)	Placebo Arm (n = 370)	HR (95% CI)
Age ≥ 65 years	48%	46%	NA
Completed treatment	177	249	NA
Discontinued (adverse event)	113	48	NA
Discontinued (progression)	24	44	NA
Deaths during or 28 days post-study treatment [(%)]	5 (1.4%) **	2 (0.5%)	NA
Non-infectious pneumonitis leading to discontinuation	2%	0.5%	NA
2-year DFS, all patients **	77.8%	77.0%	0.92 (0.67–1.22)
2-year OS, all patients	90.7%	88.3%	0.75 (0.31–1.10)
2-year LSS, all patients ***	95.2%	90.7%	0.66 (0.41–1.07)
2-year DFS, IPI ≥ 4 (n = 313) ***	82%	71%	0.65 (0.42–1.01)
2-year DFS, males (n = 372) ***	82%	75%	0.68 (0.45–1.05)
2-year DFS, Asian (n = 226) ***	83%	67%	0.60 (0.36–1.00)
2-year DFS, <65 years (n = 393) ***	80%	76%	0.79 (0.52–1.21)
2-year OS, IPI ≥ 4 (n = 313) ***	91%	82%	0.63 (0.37–1.07)
2-year OS, males (n = 372) ***	91%	86%	0.55 (0.32–0.94)
2-year OS, Asian (n = 226) ***	91%	85%	0.51 (0.27–0.98)
2-year OS, <65 years (n = 313) ***	93%	90%	0.62 (0.34–1.13)

DFS: disease-free survival, DLBCL: diffuse large B cell lymphoma, HR: hazard ratio, LSS: lymphoma-specific survival, NA: not applicable, OS: overall survival. Among 5 adverse events: related to everolimus: hepatitis B reactivation; unrelated to everolimus: congestive heart failure, septic shock, intracranial hemorrhage. ** DFS events in 23% and 27% of patients in the everolimus and placebo arms: disease progression in 20% versus 24% and deaths 3% versus 3%. *** Survival percentages were approximated from the published survival curves [87].

## Data Availability

Data sharing not applicable. No new data were created or analyzed in this study. Data sharing is not applicable to this article.

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
