# Peer review of "The Role of mTOR in B Cell Lymphoid Malignancies: Biologic and Therapeutic Aspects"

_ijms, 2023, doi:10.3390/ijms241814110_

Round 1
Reviewer 1 Report
ijms-2581317
Title: The Role of mTOR in B-cell Lymphoid Malignancies: Biologic and Therapeutic Aspects
Authors: Eleni A Karatrasoglou *, Maria Dimou, Alexia Piperidou, Eleftheria Lakiotaki, Penelope Korkolopoulou, Theodoros P. Vassilakopoulos
[Major concerns]
1. Abbreviations: The use of abbreviations when writing a paper has many advantages besides simplicity of expression. To use an abbreviation, first write the abbreviation in parentheses after the full name, and then use the abbreviation from Introduction to the final Conclusion. If an abbreviation is not used more than twice, there is no need to define it, so please delete it. In the case of an abstract, the use of abbreviations is completely separate from the main text. As you know, there are cases where only the abstract is introduced separately, so in the abstract, abbreviations should only be used if they are repeatedly used and if they are not used again, only the full name should be used.
2. The English sentences are generally well-constructed; however, there is inconsistency in the spelling of certain words and terms. The authors have meticulously distinguished between cases where it is necessary to capitalize the first letter of a disease name, gene designation, or protein nomenclature within sentences in the paper and cases where such capitalization is unnecessary. Examples: mTOR at Line 15 vs. m-TOR at Lines 21, 35, and more. Space should be placed between the final word of a sentence and the citation reference number. Not being proper nouns, correct all unnecessary capitalizations of compound names, etc. Examples: Temsirolimus and Everolimus at Lines 35 and 36.
3. In cases where abbreviations are used within figures, please list these abbreviations along with their corresponding full names in the figure legends. If there are two or more abbreviations, arrange them in alphabetical order.
4. Notation of units: In scientific papers, various units are typically separated from the preceding numerical value by a space. Some units, such as percent and temperature units, may be written together without a space, but in these cases, a consistent pattern should always be used. Examples: 250mg and 25mg at Line 191, etc.
[Minor concerns]
1. Line 76: ‘NFκB’ should be written as ‘NF-κB’.
2. Lines 120 and 123: Write the original term first, followed by the abbreviation in parentheses.
3. Line 204: ‘table 1’ should be written as ‘Table 1’.
4. Reference section: Author should consult and peruse carefully recent issues of the journal, International Journal of Molecular Sciences (IJMS), for format and style. Also double-check the abbreviations of journal names.
5. There is also several reference without page numbers and proper information. Examples: 3, 19, 25, 29, 33, 44, 56, 68, 85 etc.
Overall, the manuscript can be considered to publication after minor revision as indicated above.

ijms-2581317
Title: The Role of mTOR in B-cell Lymphoid Malignancies: Biologic and Therapeutic Aspects
Authors: Eleni A Karatrasoglou *, Maria Dimou, Alexia Piperidou, Eleftheria Lakiotaki, Penelope Korkolopoulou, Theodoros P. Vassilakopoulos
[Major concerns]
1. Abbreviations: The use of abbreviations when writing a paper has many advantages besides simplicity of expression. To use an abbreviation, first write the abbreviation in parentheses after the full name, and then use the abbreviation from Introduction to the final Conclusion. If an abbreviation is not used more than twice, there is no need to define it, so please delete it. In the case of an abstract, the use of abbreviations is completely separate from the main text. As you know, there are cases where only the abstract is introduced separately, so in the abstract, abbreviations should only be used if they are repeatedly used and if they are not used again, only the full name should be used.
2. The English sentences are generally well-constructed; however, there is inconsistency in the spelling of certain words and terms. The authors have meticulously distinguished between cases where it is necessary to capitalize the first letter of a disease name, gene designation, or protein nomenclature within sentences in the paper and cases where such capitalization is unnecessary. Examples: mTOR at Line 15 vs. m-TOR at Lines 21, 35, and more. Space should be placed between the final word of a sentence and the citation reference number. Not being proper nouns, correct all unnecessary capitalizations of compound names, etc. Examples: Temsirolimus and Everolimus at Lines 35 and 36.
3. In cases where abbreviations are used within figures, please list these abbreviations along with their corresponding full names in the figure legends. If there are two or more abbreviations, arrange them in alphabetical order.
4. Notation of units: In scientific papers, various units are typically separated from the preceding numerical value by a space. Some units, such as percent and temperature units, may be written together without a space, but in these cases, a consistent pattern should always be used. Examples: 250mg and 25mg at Line 191, etc.
[Minor concerns]
1. Line 76: ‘NFκB’ should be written as ‘NF-κB’.
2. Lines 120 and 123: Write the original term first, followed by the abbreviation in parentheses.
3. Line 204: ‘table 1’ should be written as ‘Table 1’.
4. Reference section: Author should consult and peruse carefully recent issues of the journal, International Journal of Molecular Sciences (IJMS), for format and style. Also double-check the abbreviations of journal names.
5. There is also several reference without page numbers and proper information. Examples: 3, 19, 25, 29, 33, 44, 56, 68, 85 etc.
Overall, the manuscript can be considered to publication after minor revision as indicated above.
Author Response
Thank you for your interest and your comments for the improvement of our manuscript entitled “The Role of mTOR in B-cell Lymphoid Malignancies: Biologic and Therapeutic Aspects”. The corrections are highlighted. We hope that our revised manuscript is in accordance with your requirements.
Major concerns
- The manuscript has been corrected according to your suggestions regarding abbreviations.
- The manuscript has been corrected according to your suggestions, regarding the punctuation marks, capital letters, special characters and spaces.
- We added a line at the end of each table to list abbreviations in alphabetical order. The paper does not include figures.
- Units have been separated from the preceding numerical value by a space.
Minor concerns corrected
Reviewer 2 Report
In this review, Karatrasoglou and Dimou et al. have summarized the clinical applications of m-TOR inhibitors in common B-cell lymphomas, with a specific focus on temsirolimus. Overall, the review is well-written and comprehensive. However, one aspect that appears to be missing from the review is a discussion of potential challenges and future directions. Given that the review primarily addresses the role of mTOR in B-cell malignancies, it would be beneficial to conclude by highlighting the insights from recent studies, such as PMID: 36352190, and other relevant research in this area.
Author Response
Thank you for your interest and your comments for the improvement of our manuscript entitled “The Role of mTOR in B-cell Lymphoid Malignancies: Biologic and Therapeutic Aspects”. As you suggested, in order to discuss potential challenges and future directions concerning the role of mTOR in B-cell malignancies, we added a new paragraph on lines 593-607 based on data of recent literature. The correction is highlighted. We hope that our revised manuscript is in accordance with your requirements.